# Designing multimedia patient education materials for adolescent idiopathic scoliosis: A protocol for a feasibility randomized controlled trial of patient education videos

**Garett Van Oirschot** [1]*, **Cailbhe Doherty**[1,2]

1 School of Public Health, Physiotherapy & Sport Science, University College Dublin, Dublin, Ireland,
2 Insight SFI Research Centre for Data Analytics, University College Dublin, Dublin, Ireland

* garett.vanoirschot@ucdconnect.ie

**Data Availability Statement:** No datasets were generated or analysed during the current study. All relevant data from this study will be made available upon study completion.

## Abstract

Triple-masked three-armed feasibility parallel randomized controlled trial. Multimedia patient education materials are increasingly used in healthcare. While much research focuses on optimising their scientific content, research is equally needed to optimise design and implementation. This study aims to determine the feasibility of a study examining how the implementation of scientific advice on design affects patient outcomes. Participants aged 10–18 with radiographically confirmed adolescent idiopathic scoliosis will be recruited from community settings in Ireland and randomized into usual care or receiving multimedia educational videos with or without evidence-informed design principles. Participants will be masked in the two video intervention arms, as will the therapist sending the educational videos. Outcomes will include the number of participants recruited and randomized, the number analysed post-intervention and at week eight, and the outcomes for baseline, post-intervention, and week 8. Adverse events will also be reported. This feasibility randomized controlled trial will offer insight into the feasibility of implementing advice from the literature in designing a trial of multimedia patient education materials for a population with adolescent idiopathic scoliosis.

**Trial registration**: **Clinical Trail**: Trial is registered on ClinicalTrials.gov as NCT06090344.

## Introduction

Patient education is a pillar of treatment that is recommended in many aspects of musculoskeletal (MSK) care [1–10], including adolescent idiopathic scoliosis (AIS) [11,12]. Multimedia education is one option for conveying some of the information that healthcare providers may wish to pass on to a patient. They provide the accessibility, universality and usability that is desired in patient education aids [13]. While much healthcare research focuses on the content of educational materials, there are fewer interventional studies that examine the influence of design. Where such design research does exist, it is often in other medical areas [14–27] and

**Funding:** The author(s) received no specific funding for this work.

**Competing interests:** The authors have declared that no competing interests exist.

the small amount of design recommendations in MSK healthcare are based mostly on literature arising narrative reviews [28–30], non-randomized studies, cross-sectional surveys, or case series [12]. Regardless, there is potential for educational materials to fill a gap where healthcare services are known to be under strain, which can occur due to a lack of resources or rural/remote locations [31].

Under resourcing is found in public healthcare systems for scoliosis, where waitlists can prevail for services [32–34], mirroring the dilemma of many musculoskeletal health services. Such services cannot rely on immediate one-to-one clinical encounters to provide education, whether for financial or geographical reasons [35]. Innovation has been required in healthcare to bring information and guidance to patients in these situations, with digital mediums providing such solutions. Scoliosis patients and their families have well-documented information needs [12,36,37] and multimedia education materials offer the advantage of dispersing healthcare information in a manner that is cheap, en masse [31,38,39], and without physical proximity [40,41].

### Rationale

Multimedia education has been examined extensively in the pedagogical literature, resulting in recommendations on how to best design materials in order to maximise engagement and learning [13,42] While this information has informed the design of educational materials in healthcare [14–27] this has been less frequent in the MSK literature with only one RCT examining osteoarthritis-related pain [43], and other MSK conditions being informed by narrative advice only [28–30]. The use of such principles [42] has not been tested in a randomized controlled trial of education for a MSK population such as adolescent idiopathic scoliosis.

Given the uncertainty surrounding the recruitment of this age bracket from the community and the unknown ability to engage with patient education materials over a trial period, a feasibility study is warranted to provide information on the design and implementation of a potential trial.

### Aims and objectives

The aim of the eventual three-armed parallel randomized controlled trial will be to compare the effectiveness of multimedia educational videos with and without evidence-informed design implementation against usual care for patients with adolescent idiopathic scoliosis. It will have an objective to examine if using videos with evidence-informed design results in superior post-treatment, 8 week, and 26 week outcomes for video engagement, knowledge translation and retention, and patient reported outcomes compared to videos formatted to match existing online resources [44] or treatment as usual for this population with AIS. We hypothesise that such an intervention will result in superior outcomes for video engagement, knowledge translation and retention, and for patient reported outcomes.

The more immediate aim of this feasibility RCT will be to assess the feasibility of running this trial with the objective of examining the recruiting, the intervention use, the intervention adherence, and the outcome assessment at baseline and eight weeks. It will be planned and evaluated using the RE-AIM framework [45] in order to optimise reach, effectiveness, adoption, implementation, and maintenance and determine the feasibility of this RCT and any aspects that cannot be pragmatically explored in sufficient depth will be explained to avoid shortcomings in reporting that have been previously observed [46–48].

### Materials and methods

This triple-masked three-armed feasibility parallel randomized controlled trial will be conducted in Ireland and is prospectively registered on ClinicalTrials.gov. It was designed in

consideration of the CONSORT guidelines for feasibility studies and the TIDieR [49] and RoB-2 [50] appraisal tools. The RE-AIM framework [45] will be used to ensure proper consideration of sustainable adoption and implementation. While it is not expected or intended, any changes to the trial protocol will be reported along with reasons for such changes. The trial is approved by University College Dublin Human Research Ethics Committee (LS-23-15-VanOirschot-Doherty, 3 May 2023).

## Participants

The inclusion criteria will be as follows: aged 10–18 years, parent/guardian consent for those under 18 years of age, AIS confirmed during adolescence by Cobb angle ≥10deg on plain film radiographs, able to watch and listen to online educational materials as well as read and complete online surveys. The age and 10-degree cut-off criteria were selected to ensure consistency with previous AIS literature [11,51] and international research bodies [52]. Non-AIS scoliosis including but not limited to neurological conditions will be excluded.

## Patient and public involvement

Two individuals over age 18 who have been through the adolescent care pathway for AIS were consulted when designing the study protocol for feedback on the design and educational content to be used in the interventions. Information was also sought from a physiotherapist who treats scoliosis on a daily basis and collaborates with scoliosis orthopaedic surgeons in Ireland. Their opinion will also be sought about disseminating the results, which parts of the results should be shared, and in which digital/or print format.

## Recruitment

Participants will be recruited via three private healthcare clinics serving an AIS population and three scoliosis advocacy groups based in Ireland using verbal, digital and poster advertising within these organizations alongside social media and word of mouth. The primary author, a PhD student and physiotherapist of 18 years' experience with a post-graduate musculoskeletal and sports masters degree, will conduct the preliminary screening and provide the participant information package to those who express interest. All participants will be screened by phone or by video call prior to inclusion, and all participants not meeting eligibility will have their exclusion reasons recorded but without any identifying information. Advertising and recruitment opened on 28 November 2023, with the study due to commence in early 2024.

## Consent process

Informed consent will be sought from all potential participants meeting the inclusion criteria. The primary author will discuss the trial methodology and answer any questions or concerns about the trial before taking consent / assent electronically. Those aged 18 will be able to consent for themselves, while potential participants under the age of 18 will require parent/guardian consent before granting assent themselves. All participant rights will remain protected.

## Fidelity

Treatment fidelity for the two video intervention groups will be monitored through YouTube analytics, which will show the lack of views for each participant with their video or with sections of video that are skipped over.

**Table 1. Baseline and outcome variables collected and their purpose, with the primary feasibility outcomes in bold.**

| Variable | Measure | Time points (weeks) | Purpose & dimension of RE-AIM framework |
|---|---|---|---|
| Age | Date of birth | 0 | Describe population, reach |
| Gender | Male/female | 0 | Describe population, reach |
| Date of plain film radiographs confirming scoliosis | Weeks | 0 | Describe population, reach |
| Date of last radiograph | Weeks | 0 | Describe population, reach |
| Duration since care-seeking | Weeks | 0 | Describe population, reach |
| Major Cobb Angle | Degrees | 0 | Describe population |
| Minor Cobb Angle (if applicable) | Degrees | 0 | Describe population |
| Brace treatment | Yes/No | 0 | Describe population |
| Surgical treatment | Yes/No | 0 | Describe population |
| Physiotherapy treatment | Yes/No | 0 | Describe population |
| Quality of life | SRS-22 | 0, 8 | Secondary outcome |
| | EQ-5D-Y | 0, 8 | Secondary outcome |
| Physical Activity | PAQ-C | 0, 8 | Secondary outcome |
| Anxiety | STAI-Ch | 0, 8 | Secondary outcome |
| Immediate Knowledge translation | 5-item MCQ | 1,2,3,4,5,6,8 | Secondary outcome |
| Cognitive Engagement | | | |
| *One week Knowledge retention* | 5-item MCQ | 2,3,4,5,6,7 | Secondary outcome, maintenance |
| *End of study Knowledge retention* | 30-item MCQ | 0, 8 | Secondary outcome, maintenance |
| Affective engagement | YouTube Like/Dislike | 1,2,3,4,5,6 | Secondary outcome |
| **Intervention adherence / Behavioral Engagement** | | | |
| *Watch Time* | **YouTube watch time %** | **1,2,3,4,5,6** | **Primary outcome, adoption, implementation** |
| *Viewings* | **YouTube # views** | **1,2,3,4,5,6** | **Primary outcome, adoption, implementation** |
| **Number recruited, rejected, randomized, dropped off, completed** | **Number** | **0,1,2,3,4,5,6,7,8** | **Primary outcome, implementation** |
| **Dropout reasons** | **Participant response** | **1,2,3,4,5,6,7,8** | **Primary outcome, implementation** |
| **Outcomes completed** | **%** | **8** | **Primary outcome, implementation** |
| **Adverse events** | **Number and type** | **1,2,3,4,5,6,8** | **Primary outcome, effectiveness** |
| **Satisfaction with care and treatment** | **Tailored question** | **8** | **Primary outcome, effectiveness** |

Abbreviations: SRS-22r, Scoliosis Research Society 22-revised; EQ-5D-Y, EuroQOL 5 Dimension Youth version; PAQ-C, Physical Activity Questionnaire for Children; STAI-Ch, State Trait Anxiety Index for Children; MCQ, Multiple Choice Questionnaire.

## Baseline assessment

Following informed consent, participants will electronically complete a series of baseline assessments of demographics and outcome measures, with the primary author being available by phone or electronically if assistance is required. A description of the baseline variables plus other study outcomes and their purpose is shown in Table 1.

## Randomization & masking

Upon consenting to participate in the study, participants will be randomized by a random number generator on a 1:1:1 ratio into one of the three intervention arms by a member of the research team who will be masked to the interventions and follow-up assessment of outcomes. Participants will be told that the trial will compare usual care with two other multimedia video interventions. Participants in the usual care group will not be masked to their allocation but the participants in the two video interventions will not be aware of their allocation to videos with traditional format or augmented format.

For the two video groups, an email will be sent by the supervising author, a registered Physiotherapist and researcher for fifteen years, with a link to the correct video but this author will be masked to which intervention the link pertains to. The primary author will record and analyze all outcomes that will be taken via the SurveyMonkey online platform but will remain masked to the treatment delivered when viewing these outcomes.

## Interventions

In the usual care group, participants will continue along their existing healthcare pathway and will not receive any educational intervention beyond the usual education from their healthcare providers, which will not be determined by participation in the study and will not be funded by it.

In the video groups, participants will continue along their usual healthcare pathway as well, however they will also be sent a link to six multimedia videos, each three to four minutes long, using the information and format of a typical online resource, in this case the Frequently Asked Questions section of Scoliosis Research Society website [44] with some additional content from the qualitative literature on information needs in the scoliosis population [11,36,37,53]. The outline of the content is shown in Table 2. Adjustments were also made to ensure spelling and grammar in Irish-version English language and some terms were altered to reflect Irish phrases i.e. consultant instead of specialist. The script was edited again following feedback from the individuals who formed the public and patient involvement described above, and this assisted in making minor corrections and updating statistics such as higher prevalence in girls vs boys [54–57], as well as elaborating on discussion points about lifestyle factors and back pain. The video channel will be set up as a private channel unique to each participant, so that the engagement statistics can be tracked for each participant. This will consist of six videos, sent once weekly from Week 1 to Week 6. Participants can skip forward, back and re-watch a released video an unlimited number of times until Week 8.

**Similarities between the two video intervention groups.** The videos will contain identical information and it will be presented in the same sequence in both types of videos. The script will be identical except for specific components that are described below.

**Differences between the two video intervention groups.** For the video group with augmented (AUGM), the text, graphics, and audio formatted in accordance with recommendations from the Cognitive Theory of Multimedia Learning and other advice from the literature [13,28–30,42].

## Materials

The multimedia materials for the TRAD and AUGM video groups will be recorded using an iPhone13 to allow for content to be formatted to a phone screen format in case users view it on mobile devices and edited on Final Cut Pro X [58] with additional audio editing on Garage Band [59]. The final video will upload to a private YouTube channel setup by the primary author and the corresponding link will be sent via email. All participants will require internet access and a device to view and/or hear the educational materials. Participants in the two video intervention groups will be sent a link to a private YouTube channel, unique to each participant so that individual YouTube analytics can recorded and analyzed. Additional patient-reported outcomes will be collected via SurveyMonkey.

Following completion of the study, the videos will be made available on a YouTube link for all participants. They will also be made public via an open link in the final manuscript of this study, to assist with any future replication and appraisal.

**Table 2. Outline of educational content.**

| Video | Questions to be answered in the educational content |
|---|---|
| 1 | What is scoliosis?<br>Is scoliosis more common in boys or girls?<br>What are the other types of scoliosis?<br>Does everyone with scoliosis wear a brace or need surgery?<br>Why do kids get scoliosis?<br>Does bad posture lead to scoliosis?<br>Are there exercises I can do to make my spine straight? |
| 2 | Can we screen for scoliosis?<br>Can scoliosis curves get better on their own?<br>How often will I need my scoliosis re-checked?<br>Will the x-rays harm me? |
| 3 | What can I do to keep my scoliosis from getting worse?<br>How do you decide what the treatment will be?<br>What is involved in conservative?<br>Can you tell me about bracing?<br>How do you know if a patient needs surgery?<br>Where can I find healthcare specialists who work with scoliosis?<br>Should I try other conservative methods first? |
| 4 | Which kind of brace is right for me?<br>What are the different types of braces?<br>Why should I wear a brace?<br>Does it matter how many hours a day I wear the brace?<br>Can I take the brace off for PE at school, sports activities, swimming, etc.?<br>How will I know if the brace is working?<br>I wore a brace for two years and my curve was the same size when I stopped wearing it, why? Did I waste my time?<br>What does a brace look like? |
| 5 | If I need scoliosis surgery, what can I expect?<br>Will surgery only stop my curve from growing or will it help straighten it?<br>How long will it take me to recover from surgery?<br>Does that include participation in sports?<br>Will I have my rods removed after my spine is fused?<br>Can the rods break? |
| 6 | Should I avoid certain activities because of my scoliosis?<br>Is carrying a heavy back-pack bad?<br>What if I have to wear a brace?<br>How do I cope with having scoliosis?<br>How can I connect with other kids who have scoliosis?<br>Will my curves get worse?<br>Can my scoliosis be cured?<br>If I have mild scoliosis, do I have to see a spine specialist?<br>Will I pass my scoliosis on to my future children? |

## Outcomes

As recommended in the CONSORT guidelines for feasibility studies [60,61], the feasibility outcomes for this trial will be the number of participants recruited, rejected, dropped out and reasons; adherence to videos via behavioural engagement; number of outcomes completed, and adverse events. Adverse events will be recorded with preparations made for any adverse effects from asking about the participants' mental health status or from being presented with unexpected information about scoliosis. The preparations are outlined Table 3.

Adherence to the RE-AIM framework will also ensure that there is an evaluation of the reach: recording of the percentage of excluded individuals, the percentage of individuals who participate, the characteristics of participants versus non-participants and the recording of qualitative indicators as to why participants were reached and/or recruited. The effectiveness will be measured using the knowledge outcomes and the overall quality of life outcomes in the SRS-22r

**Table 3. Protocol for adverse events.**

| Potential event | Research member responsible | Action to be taken |
|---|---|---|
| Participant distress from answering mental health questions | Primary author will be available by phone or electronic means | Provide support and explain the rationale for the questions in the outcome measures and if needed, liaising with a psychologist with experience in treating an AIS population. |
| Participant distress from receiving unexpected information about AIS | Primary author will be available by phone or electronic means | Answer any AIS-related questions and provide reassurance where needed. Onward referral can be made back to any scoliosis-specialised health services that the participant is already attending, or to a healthcare practitioner known to the primary author who has experience with this population |

and the EQ-5D-Y, a moderation analysis across the subgroups, and attrition analysis of the characteristics of these participants and their reasons. Adoption to specific settings is expected to be assessed easily by the online delivery of this intervention. The implementation will be evaluated by recording the number of interventions delivered perfectly, the adaptations made over the course of the study, the total budget, and the consistency of implementation across the eight weeks. Maintenance will be assessed by the measurement of all outcomes at the follow-up assessment in week eight, attrition rate, and analysis of the subgroups for each of these items.

The final fully powered RCT, should it proceed based on results of this feasibility study, will assess a primary outcome of post-intervention cognitive engagement, meaning the knowledge translation for each video for weeks 1–6 and retention for all videos at Week 8. Secondary outcomes will be knowledge retention one week later for each of the six videos for weeks 2–7; affective & behavioural engagement from YouTube analytics determined by like/dislike, watch-time, number of views; health related quality of life measured by the SRS-22r and the EQ-5D-Y, anxiety measured by the STAI-Ch, and physical activity measured by the PAQ-C. Secondary outcomes will be measured at baseline and week 8.

The SPIRIT schedule of enrolment, interventions, and assessments is shown in Fig 1.

## Statistical analysis

SPSS version 27.0 for Mac (SPSS, IBM Inc., Chicago, IL) will be used to analyse study data. Ratio and interval variables (YouTube analytics) will be analysed for differences by ANOVA. The remaining ordinal variables will have differences between groups analysed using Friedmans. Pre- and post-test changes will be analysed using Kruskal-Wallis. Comparisons will be made using intention-to-treat principles.

Generalised estimating equations (GEE) will be used to evaluate the relationship between the design characteristics and the measures of engagement. Each of the characteristics described above (Watch time / Total time (%), Retention (%)), Time spent on specific segments, Mean view duration (minutes:seconds) will be included as co-variates in separate GEE models for each measure of engagement.

There was an audit of previous year's records to note that were approximately 60 eligible participants over twelve months in one of the private practice recruitment sites, and this will be in addition to social media and advocacy groups advertising (with two advocacy groups having 2542 and 579 followers in Ireland), as well as two other pending consultant clinics. Statistical power of 80% for the eventual RCT has been calculated using GPower [62] to detect a minimum change of 1 out of 5 in the primary outcome, knowledge quizzes, with a one-way ANOVA to compare three groups with an alpha of 0.05 and determined that a sample size of

| | STUDY PERIOD | | | | | | | | | | |
|---|---|---|---|---|---|---|---|---|---|---|---|
| | Enrolment | Allocation | Post-allocation | | | | | | | | Close-out |
| TIMEPOINT | *-8 wk* | *0* | *1 wk* | *2 wk* | *3 wk* | *4 wk* | *5 wk* | *6 wk* | *7 wk* | | *8 wk* |
| **ENROLMENT:** | | | | | | | | | | | |
| **Eligibility screen** | X | | | | | | | | | | |
| **Informed consent** | X | | | | | | | | | | |
| **Allocation** | | X | | | | | | | | | |
| **INTERVENTIONS:** | | | | | | | | | | | |
| *Augmented Videos* | | | ◆——————————————————◆ | | | | | | | | |
| *Traditional Videos* | | | ◆——————————————————◆ | | | | | | | | |
| *Usual care* | | │ | ◆——————————————————◆ | | | | | | | | |
| **ASSESSMENTS:** | | | | | | | | | | | |
| *Baseline variables* | | X | | | | | | | | | |
| *SRS-22r, EQ-5D-Y, STAI-Ch, PAQ-C* | | X | | | | | | | | | X |
| *Engagement* | | | X | X | X | X | X | X | X | | X |
| *Knowledge (weekly topics)* | | | X | X | X | X | X | X | X | | |
| *Knowledge (all topics)* | | X | | | | | | | | | X |

Note: wk = Week

**Fig 1. SPIRIT schedule of enrolment, interventions, and assessments.**

92 participants were needed. This feasibility study will assess the recruitment potential using 10–15 participants per group.

Data will be stored and handled using password-protected institutional storage services, in accordance with institutional policy. The final data will be accessibly by the primary and supervising authors.

## Results

This trial will be reported in conformance with the CONSORT guideline for feasibility randomized controlled trials. A CONSORT flow diagram will outline the number of participants in each stage of the study (Fig 2). All anonymised data will be publicly available as per Open Science recommendations on https://osf.io/rcfsd/. Results will be communicated to participants directly and through academic publication with no publishing restrictions.

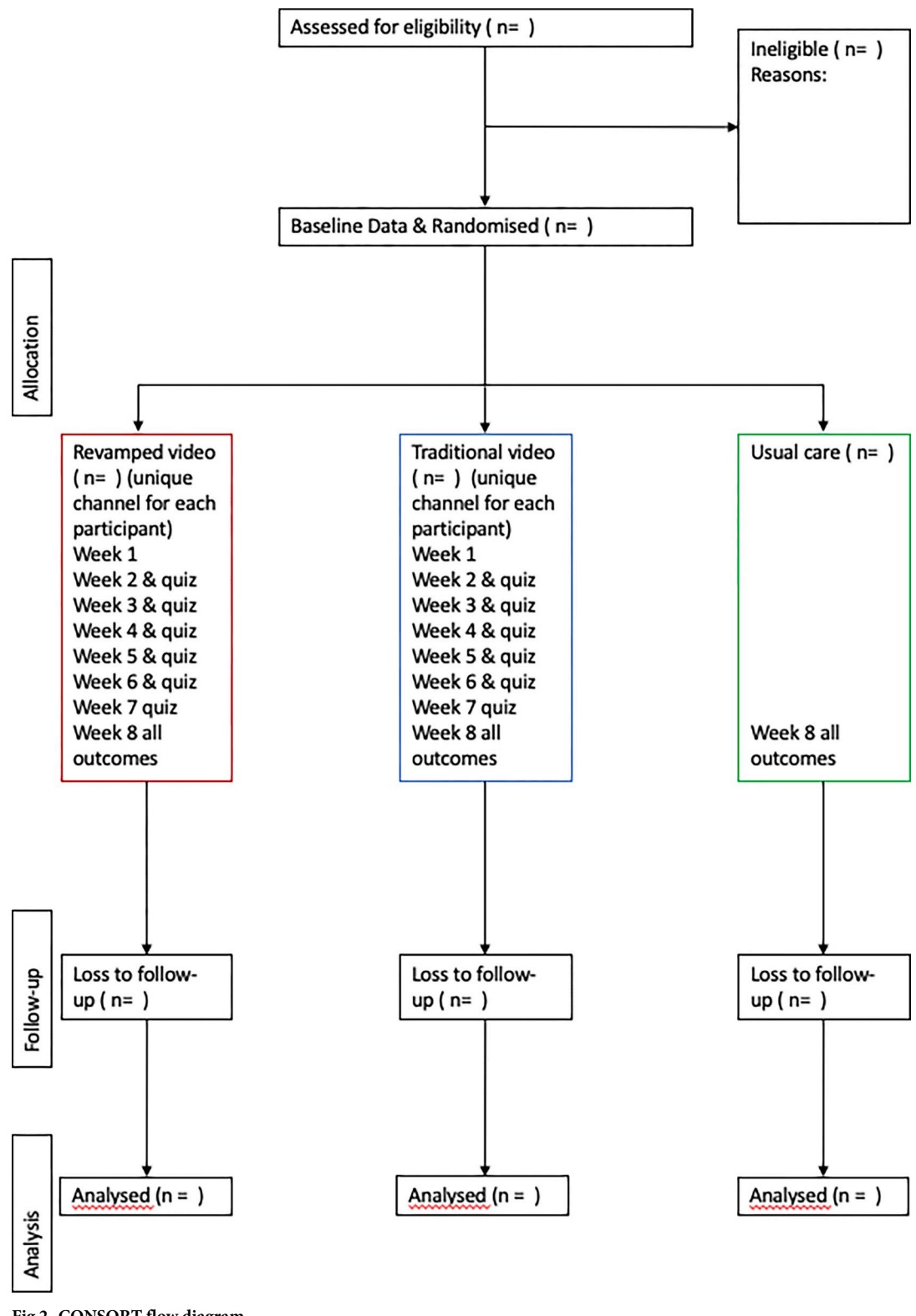

**Fig 2. CONSORT flow diagram.**

## Discussion

### Potential limitations and attempts to address

SRS-22r does not capture all aspects of physical functioning, such as mobility and self-care [63]. Additionally, SRS-22r was evaluated in the COS study that included all spinal deformities and thus limiting applicability to specific AIS [63] therefor additional quality of life measure will be taken via the EQ-5D-Y. It is acknowledged the EQ-5D-Y may require supplemental assessment tools for school function [64], and that it's correlation with the SRS-22r has been found to be good for non-scoliotic populations [65]. Proxy reporting may occur with the EQ-5D-Y as participants complete at home in the presence of parent(s)/guardian(s), but this yields similar outcomes as the 3-layer predecessor [66].

### Future potential

This study will determine the feasibility of implementing a fully powered trial and may offer an option to provide basic information about AIS to patients and the general public. Its utility could be particularly found in addressing those individuals on the waitlists for many services related to AIS [11,32,33]. This could have further implications for countries with under resourced healthcare systems or rural and remote regions lacking the access to clinicians that would often provide this specialty advice and education.

## Conclusion

This feasibility randomized controlled trial will offer insight into the feasibility of implementing advice from the literature in designing multimedia patient education materials for a population with adolescent idiopathic scoliosis and allow a comparison with materials that do not undergo similar redesign as well as a comparison with usual care.

## Supporting information

**S1 Checklist. SPIRIT 2013 checklist: Recommended items to address in a clinical trial protocol and related documents\*.**
(DOC)

**S1 File.**
(PDF)

## Acknowledgments

The authors wish to thank Scoliosis Awareness & Support Ireland, the Scoliosis Advocacy Network, Scoliosis Ireland, McGowan Physio, Orthotic Solutions, and Activ8 Physio for their offer of assistance in recruiting for this study.

## Author Contributions

**Conceptualization:** Garett Van Oirschot, Cailbhe Doherty.

**Data curation:** Garett Van Oirschot, Cailbhe Doherty.

**Formal analysis:** Garett Van Oirschot, Cailbhe Doherty.

**Funding acquisition:** Garett Van Oirschot.

**Investigation:** Garett Van Oirschot, Cailbhe Doherty.

**Methodology:** Garett Van Oirschot, Cailbhe Doherty.

**Project administration:** Garett Van Oirschot, Cailbhe Doherty.

**Resources:** Garett Van Oirschot, Cailbhe Doherty.

**Software:** Garett Van Oirschot.

**Supervision:** Cailbhe Doherty.

**Validation:** Garett Van Oirschot.

**Visualization:** Garett Van Oirschot.

**Writing – original draft:** Garett Van Oirschot.

**Writing – review & editing:** Garett Van Oirschot, Cailbhe Doherty.

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
