## [Decision Letter · Decision Letter 0]

6 Mar 2024

PONE-D-23-39588Designing multimedia patient education materials for adolescent idiopathic scoliosis: protocol for a feasibility randomized controlled trial of patient education videosPLOS ONE

Dear Dr. Van Oirschot,

Thank you for submitting your manuscript to PLOS ONE. After careful consideration, we feel that it has merit but does not fully meet PLOS ONE’s publication criteria as it currently stands. Therefore, we invite you to submit a revised version of the manuscript that addresses the points raised during the review process.

We look forward to receiving your revised manuscript.

Kind regards,

Holakoo Mohsenifar

Academic Editor

PLOS ONE

Journal Requirements:

Reviewers' comments:

Reviewer's Responses to Questions

**Comments to the Author**

1. Does the manuscript provide a valid rationale for the proposed study, with clearly identified and justified research questions?

Reviewer #1: Yes

Reviewer #2: Partly

2. Is the protocol technically sound and planned in a manner that will lead to a meaningful outcome and allow testing the stated hypotheses?

Reviewer #1: Yes

Reviewer #2: Yes

3. Is the methodology feasible and described in sufficient detail to allow the work to be replicable?

Reviewer #1: Yes

Reviewer #2: Yes

4. Have the authors described where all data underlying the findings will be made available when the study is complete?

Reviewer #1: No

Reviewer #2: Yes

5. Is the manuscript presented in an intelligible fashion and written in standard English?

Reviewer #1: Yes

Reviewer #2: Yes

6. Review Comments to the Author

You may also provide optional suggestions and comments to authors that they might find helpful in planning their study.

Reviewer #1: In this study protocol, a three-arm feasibility randomized control trial is accruing which aims to determine the feasibility of examining how the implementation of scientific advice on design affects patient outcomes.

Minor revisions:

1- The standard statistical term for average is mean.

2- Page 18: State the power of the ANOVA test with n=92.

3- Indicate the statistical software that will be used for the analysis.

4- To assist in the review process, please add line numbers to the document.

Reviewer #2: Dear Authors,

I have thoroughly reviewed your manuscript submitted to PLOS ONE, titled "Designing multimedia patient education materials for adolescent idiopathic scoliosis: protocol for a feasibility randomized controlled trial of patient education videos". Based on my assessment, I provide the following feedback for your consideration:

Exclusion Criteria: I recommend revising the exclusion criteria to explicitly exclude individuals who have undergone any previous treatments for scoliosis. This will help in minimizing variability that could confound the study outcomes.

Inclusion Criteria Regarding Cobb Angle: The manuscript considers including patients with a Cobb angle greater than 10 degrees. However, clinical guidelines generally recommend interventions for curves exceeding 20 degrees. Aligning your inclusion criteria with these guidelines may enhance the clinical relevance of your findings.

Age Range of Participants: The current age range for participants is set between 10 to 18 years. Given that adolescence typically ends around 15 or 16 years due to skeletal maturity, I suggest adjusting this range to more accurately reflect adolescent scoliosis, thereby avoiding the inclusion of adult scoliosis cases.

Recruitment Methods: The recruitment strategy involving social media and word of mouth might not ensure the targeted and rigorous approach necessary for this study. A more structured recruitment method, perhaps involving referrals from healthcare professionals, is advisable.

Study Timeline: It is crucial to update all study materials to accurately reflect the current timeline, especially considering the mentioned start date of January 2024.

While I appreciate the effort and dedication behind your manuscript, these recommendations aim to enhance its scientific rigor and alignment with clinical practices.

7. PLOS authors have the option to publish the peer review history of their article (what does this mean?). If published, this will include your full peer review and any attached files.

Reviewer #1: No

Reviewer #2: **Yes: **Mojtaba Kamyab

---

## [Author Response · Author response to Decision Letter 0]

13 Apr 2024

13 April 2024

Dear Academic Editor & reviewers, 

Thank you kindly for your time in reviewing our manuscript “Designing multimedia patient education materials for adolescent idiopathic scoliosis: protocol for a feasibility randomized controlled trial of patient education videos” and providing your valuable feedback. 

Your suggestions have been helpful in amending our manuscript and clarifying many aspects of this study protocol. For each of your suggestions, we have responded directly below. 

Reviewer 1: 

The standard statistical term for average is mean.

Thank you for this edit. It has now been addressed on p16 line 284. 

Page 18: State the power of the ANOVA test with n=92.

 Thank you again for this suggestion. It has been more clearly stated on p16 line 291.

Indicate the statistical software that will be used for the analysis.

 We have clarified the use of SPSS for Mac on p16 line 275-6. 

To assist in the review process, please add line numbers to the document.

 We apologise for neglecting this on the original draft of the manuscript. Please find line numbers added to the revised version. 

Reviewer 2: 

Exclusion Criteria: I recommend revising the exclusion criteria to explicitly exclude individuals who have undergone any previous treatments for scoliosis. This will help in minimizing variability that could confound the study outcomes.

 Thank you for the feedback on this aspect of our recruitment. There were discussions surrounding this during the study design phase , but it was found to not be feasible because

- based on researcher experience and discussions with potential recruitment sites, diagnosis and initiation of treatment planning will typically be underway by the time participants can be recruited, and is also likely to have commenced for most participants who are in contact with charity & advocacy groups acting as additional recruitment sources 

- even if the previous concerns were overcome, the relatively small population of the Republic of Ireland and the time constraints associated with the study, create a situation where recruitment numbers would not be met if the study was limited to only those who have not started any treatment for their scoliosis

Our additional rationale for allowing participants who have received treatment to be included

- Considering this will be a feasibility study, the primary outcomes of number recruited, dropped out, followed-up, and the reasons for dropout are the most pertinent outcomes for determining the feasibility of such a trial 

- Studying participants from various points along a treatment pathway offers the opportunity to examine if and when such an educational intervention is most appropriate, another benefit to such a feasibility study 

- Upon completing this trial and a determination that it is feasible, then the necessary time and funding can be sought to conduct the full scale trial where sufficient time and resources can be allotted to recruit the desired number of participants with no prior treatment or who are just starting their treatment 

Inclusion Criteria Regarding Cobb Angle: The manuscript considers including patients with a Cobb angle greater than 10 degrees. However, clinical guidelines generally recommend interventions for curves exceeding 20 degrees. Aligning your inclusion criteria with these guidelines may enhance the clinical relevance of your findings.

Thank you for noting the Cobb Angle requirements and for sharing your expertise on this part of our study. We are in complete agreement that some guidelines will recommend certain interventions when exceeding 20 degrees. The 10 degree criteria, while below the typical threshold for bracing or surgical intervention, was determined because 

- it is beyond the amount of normal spinal curvature as defined by the Scoliosis Research Society (SRS) [1] and International Society on Scoliosis Orthopaedic and Rehabilitation Treatment (SOSORT) [2], and is in agreement with the definitions provided in most scoliosis publications and this has been explained better in the manuscript on p7 line 123-5. 

- it is within the range recommended by SRS and SOSORT for continued orthopaedic monitoring [2] and patients under such orthopaedic monitoring can benefit from scoliosis educational materials as used in this study

- it is within the range recommended by SOSORT to benefit from scoliosis-specific exercise e [2-9] and thus benefit from any educational materials surrounding this intervention 

- while Cobb angle correction may not be a prioritised goal in patient with curves under 20 degrees, pain and pulmonary function may still require healthcare intervention [7] and thus educational materials may be of benefit 

- it was advocated during the discussions with healthcare workers as part of the public and patient involvement in the design phase of the study 

Age Range of Participants: The current age range for participants is set between 10 to 18 years. Given that adolescence typically ends around 15 or 16 years due to skeletal maturity, I suggest adjusting this range to more accurately reflect adolescent scoliosis, thereby avoiding the inclusion of adult scoliosis cases.

 Thank you again for your suggestion on this. We fully agree and have amended the protocol on p6 line121 to clarify that only those diagnosed in adolescence will be included. 

Recruitment Methods: The recruitment strategy involving social media and word of mouth might not ensure the targeted and rigorous approach necessary for this study. A more structured recruitment method, perhaps involving referrals from healthcare professionals, is advisable.

 We appreciate your highlighting of this part of our protocol, which we realise was not described in sufficient detail. We have now endeavoured to be clearer about the healthcare recruitment sites intended to form the majority of our recruitment strategy on p7 line 138-142. 

Study Timeline: It is crucial to update all study materials to accurately reflect the current timeline, especially considering the mentioned start date of January 2024.

 Thank you for your comment on this, we are in full agreement and are amending all materials. 

We wish to thank you again for your feedback and for enhancing the quality of this manuscript. If any further information or clarification is needed, we would be happy to provide it. 

Kindest regards, 

The authors 

References:

- 1. Scoliosis Research Society. Scoliosis. What is Scoliosis 2023 [cited 2024 12 April]. Available from: www.srs.org/Patients/Conditions/Scoliosis.

- 2. Negrini S, Donzelli S, Aulisa AG, Czaprowski D, Schreiber S, de Mauroy JC, et al. 2016 SOSORT guidelines: orthopaedic and rehabilitation treatment of idiopathic scoliosis during growth. Scoliosis Spinal Disord. 2018;13:3. Epub 20180110. doi: 10.1186/s13013-017-0145-8. PubMed PMID: 29435499; PubMed Central PMCID: PMCPMC5795289.

- 3. Kuru T, Yeldan İ, Dereli EE, Özdinçler AR, Dikici F, Çolak İ. The efficacy of three-dimensional Schroth exercises in adolescent idiopathic scoliosis: a randomised controlled clinical trial. Clin Rehabil. 2016;30(2):181-90. Epub 20150316. doi: 10.1177/0269215515575745. PubMed PMID: 25780260.

- 4. Negrini S, Fusco C, Minozzi S, Atanasio S, Zaina F, Romano M. Exercises reduce the progression rate of adolescent idiopathic scoliosis: results of a comprehensive systematic review of the literature. Disabil Rehabil. 2008;30(10):772-85. doi: 10.1080/09638280801889568. PubMed PMID: 18432435.

- 5. Park JH, Jeon HS, Park HW. Effects of the Schroth exercise on idiopathic scoliosis: a meta-analysis. Eur J Phys Rehabil Med. 2018;54(3):440-9. Epub 20171002. doi: 10.23736/s1973-9087.17.04461-6. PubMed PMID: 28976171.

- 6. Zapata KA, Sucato DJ, Jo CH. Physical Therapy Scoliosis-Specific Exercises May Reduce Curve Progression in Mild Adolescent Idiopathic Scoliosis Curves. Pediatr Phys Ther. 2019;31(3):280-5. doi: 10.1097/pep.0000000000000621. PubMed PMID: 31220013.

- 7. Weiss HR. Rehabilitation of adolescent patients with scoliosis--what do we know? A review of the literature. Pediatr Rehabil. 2003;6(3-4):183-94. doi: 10.1080/13638490310001636790. PubMed PMID: 14713584.

- 8. Schreiber S, Parent EC, Khodayari Moez E, Hedden DM, Hill DL, Moreau M, et al. Schroth Physiotherapeutic Scoliosis-Specific Exercises Added to the Standard of Care Lead to Better Cobb Angle Outcomes in Adolescents with Idiopathic Scoliosis - an Assessor and Statistician Blinded Randomized Controlled Trial. PLoS One. 2016;11(12):e0168746. Epub 20161229. doi: 10.1371/journal.pone.0168746. PubMed PMID: 28033399; PubMed Central PMCID: PMCPMC5198985.

- 9. Karavidas N, Iakovidis P, Chatziprodromidou I, Lytras D, Kasimis K, Kyrkousis A, et al. Physiotherapeutic Scoliosis-Specific Exercises (PSSE-Schroth) can reduce the risk for progression during early growth in curves below 25°: prospective control study. Eur J Phys Rehabil Med. 2024;60(2):331-9. Epub 20240319. doi: 10.23736/s1973-9087.24.08177-2. PubMed PMID: 38502554.

---

## [Editor Report · Decision Letter 1]

29 Apr 2024

Designing multimedia patient education materials for adolescent idiopathic scoliosis: protocol for a feasibility randomized controlled trial of patient education videos

PONE-D-23-39588R1

Dear Dr. Garett Van Oirschot,

We’re pleased to inform you that your manuscript has been judged scientifically suitable for publication and will be formally accepted for publication once it meets all outstanding technical requirements.

Kind regards,

Holakoo Mohsenifar

Academic Editor

PLOS ONE
---

## [Editor Report · Acceptance letter]

13 May 2024

PONE-D-23-39588R1 

PLOS ONE

Dear Dr. Van Oirschot, 

I'm pleased to inform you that your manuscript has been deemed suitable for publication in PLOS ONE. Congratulations! Your manuscript is now being handed over to our production team.

Kind regards, 

on behalf of

Dr. Holakoo Mohsenifar 

Academic Editor

PLOS ONE